# Management and outcome of post-myocardial infarction ventricular septal rupture—A Low-Middle-Income Country Experience

Saba Aijaz[1,2]*, Ghazal Peerwani[1], Asadullah Bugti[2], Sana Sheikh[1], Marium Mustaqeem[2], Sharwan Bhuro Mal[2], Ahson Memon[3], Ghufranullah Khan[3], Asad Pathan[1,2]

1 Department of Clinical Research Cardiology, Tabba Heart Institute, Karachi, Pakistan, 2 Department of Clinical Cardiology, Tabba Heart Institute, Karachi, Pakistan, 3 Department of Cardiac Surgery, Tabba Heart Institute, Karachi, Pakistan

* sabaxlent@gmail.com, saba.aijaz@tabbaheart.org

**Data Availability Statement:** Data cannot be shared publicly because of confidentiality and also because it is a part of database. Data will be available from the Tabba Heart Institute Ethics

## Abstract

### Background

Ventricular septal rupture (VSR) is a rare complication after acute myocardial infarction (AMI) especially in the reperfusion era but its associated mortality has remained high. This case series evaluated in-hospital and intermediate-term mortality in VSR patients. Additionally, we also analyzed risk factors, clinical presentation, intervention, and predictors of in-hospital mortality in VSR patients.

### Methods

Data of 67 patients with echocardiography confirmed diagnosis of VSR from January 2011 to April 2020 was extracted from hospital medical records. Records were also reviewed to document 30 day and 1-year mortality, recurrent heart failure admission, repeat myocardial infarction, and revascularization. In addition, telephonic follow-up was done to assess health-related quality of life(HRQOL) assessed by KCCQ-12. SCAI shock classification was used to categorize severity of cardiogenic shock. Univariate and multivariable logistic regression was used to determine predictors of in-hospital mortality. Survival function was presented using the Kaplan-Meier survival curve.

### Results

Mean age of patients was 62.7 ± 11.1 years, 62.7% were males. 65.7% of the patients presented more than 24 hours after MI and did not receive reperfusion therapy. Median time from AMI to VSR diagnosis was 2 (1–5) days. VSR closure was attempted in 53.7% patients. In-hospital mortality was 65.7%. At univariate level, predictors of in-hospital mortality were non-surgical management, basal VSR, right ventricular dysfunction, early VSR post-MI, and severe cardiogenic shock at admission (class C, D, or E). Adjusted predictors

committee (contact via email: irb@tabbaheart.org) on reasonable request for researchers who meet the criteria for access to confidential data.

**Funding:** The author(s) received no specific funding for this work.

**Competing interests:** The authors have declared that no competing interests exist.

**Abbreviations:** VSR, Ventricular septal rupture; AMI, Acute myocardial infarction'; MI, Myocardial infarction; STEMI, ST-elevation myocardial infarction; PCI, Percutaneous coronary intervention; KCCQ-12, Kansas City Cardiomyopathy Questionnaire; HRQOL, Health-related quality of life; QOL, Quality of life; PASP, Pulmonary artery systolic pressure; RV, Right ventricular; LV, Left ventricular; IABP, Intra aortic balloon pump; IQR, Interquartile range; NYHA, New York Heart Association; CABG, Coronary artery Bypass grafting; CI, Confidence interval; OR, Odds ratio; IR, incidence rate; KM, Kaplan Meier; FFI, Fried Frailty Index; LMIC, Low-Middle-Income Country; GUSTO-1, Global Utilization of Streptokinase and TPA for Occluded Coronary Arteries; MCS, mechanical circulatory support; V-A ECMO, veno-arterial extracorporeal membrane oxygenation; LVAD, left ventricular assist device; ECG, Electrocardiogram.

of in-hospital mortality included non-surgical management, basal VSR and advanced cardiogenic shock. There were 5 deaths during median followup of 44.1 months. HRQOL in patients available on followup was good (54.5%) or excellent (45.5%).

## Conclusion

High in-hospital mortality was seen in VSR patients. VSR closure is the preferred treatment to get long-term survival, however, timing of repair as well as severity of cardiogenic shock plays a significant role in determining prognosis.

## 1. Introduction

Ventricular septal rupture (VSR) is a well-recognized mechanical complication of acute ST-elevation myocardial infarction (STEMI) associated with very high mortality [1–3]. The incidence appears to have decreased with the use of acute reperfusion therapy from 1% to 2% in the pre-thrombolytic era to 0.2% currently with acute STEMI percutaneous coronary intervention (PCI) [1, 2, 4]. There has been a steady trend toward earlier identification of VSR after STEMI in contemporary literature. While traditionally thought to occur between days 3–5 after MI, initial observations from the SHOCK Registry challenged this paradigm, and the median duration of VSR presentation among 55 patients was estimated at 16 h [5].

There are several independent risk factors for developing VSR in patients presenting with acute coronary syndromes, including older age, female gender, prior stroke, chronic kidney disease, and presence of acute heart failure [5–7]. In addition, patients who develop VSR are more likely to present with ST-segment elevation, elevated cardiac biomarkers at the time of presentation, cardiogenic shock, cardiac arrest, higher Killip class, and longer symptom duration prior to hospital presentation [8, 9]. Interestingly patients who develop VSR are less likely to have a history of hypertension, diabetes, prior smoking, or prior MI. These patients may have pre-existing coronary artery disease, leading to the development of protective collateral circulation.

The outcome after septal rupture was extremely poor in the pre-thrombolytic era, with a hospital mortality rate of approximately 45% in surgically treated patients and 90% in medically managed patients [1–3]. Predictors of a poor late outcome in this population included cardiogenic shock, inferior infarction, and poor right ventricular function [4, 10–12].

The objective of this VSR case series was an analysis of patients presenting with VSR, their risk factors, clinical presentation, intervention, and hospital outcomes. Additionally, we want to evaluate any clinical factors that may affect patient mortality and their follow-up outcomes.

## 2. Materials and methods

### 2.1. Study design

This is a retrospective case series of patients with echocardiography confirmed diagnosis of VSR after MI, presenting to a single institution from January 2011 to April 2020.

### 2.2. Data collection procedure

Electronic and paper medical records of patients were reviewed. The data comprised of demographic and clinical characteristics of the patient, past medical history, admission characteristics, in-hospital therapies, and events. Hospital records were also reviewed to record a 30 day

and yearly follow-up on mortality, recurrent heart failure admission, repeat myocardial infarction, and revascularization.

Kansas City Cardiomyopathy Questionnaire (KCCQ-12) was administered to the patients who were alive via telephonic call after verbal consent to determine their health-related quality of life (HRQOL). KCCQ-12 consists of 4 subscales, including physical limitations, symptoms, quality of life (QOL), and social limitations. Each item on these subscales was scored on a Likert scale [13]. The summary score of each subscale was calculated by subtracting the minimum from the maximum subscale score and then dividing it by the range of subscale. The overall HRQOL score was the arithmetic mean of all subscales scores. A score of 0-<25 indicated poor HRQOL, 25-<50 fair HRQOL, 50-<75 good HRQOL, and 75–100 excellent HRQOL [13]. Literature has quoted KCCQ-12 as a reliable tool with Cronbach's alpha >0.76 [13]. This tool has been previously used in patients with AMI [14].

Anonymized data was extracted from EMR. Institutional Review Board of Tabba Heart Institute reviewed the study waived the requirement of consent from deceased patients' family members and ethical exemption for EMR review was given. Approval for verbal consent was granted by IRB and verbal consent was taken and documented by trained medical officer during telephonic QoL interview.

### 2.3. Statistical analysis

STATA version 16 was used for statistical analysis. Continuous variables were summarized as mean±SD or Median (IQR) and categorical variables as percentages. Categorical variables were compared by the chi-square test. Continuous variables were compared using the Independent sample T-test or Mann Whitney U test. Univariate and multivariable logistic regression was used to determine predictors of in-hospital mortality. A subgroup analysis was done to determine predictors of in-hospital mortality in patients undergoing surgical treatment. Crude odds ratios were reported along with 95% confidence intervals. P-values of univariate and multivariable analysis are reported. A p-value of less than 0.05 was considered significant.

Survival function was presented using Kaplan-Meier survival curve.

## 3. Results

Over a 10 year period, out of 11,428 patients presenting with STEMI, 67 (0.6%) had definite evidence of VSR (Fig 1).

The mean age was 62.7±11.1 years, and 42 (62.7%) were males. Other characteristics are tabulated in Table 1. The majority, (n = 40, 59.7%) of patients had an anterior location of MI. Two-thirds (n = 44, 65.7%) presented more than 24 hours after MI and did not receive any reperfusion therapy. Thrombolysis was administered in 16 (23.9%), and 7 (10.4%) underwent immediate PCI. The median time from MI diagnosis to VSR diagnosis was 2 days (IQR 1–5). All patients had a systolic murmur on examination. Mean SBP was 104.9 ± 18 mmHg. The mean heart rate was 104 ± 17 per minute. At the time of VSR diagnosis, 52 (77.6%) had advanced shock (stage C, D, or E) as per SCAI shock classification [15]. IABP was inserted in 31 (46.3%) patients.

### 3.1. Echo and angiography findings

The VSR was anteriorly located in the majority (n = 40, 59.7%) of the patients and 19 (28.3%) patients had basal VSR while rest of 8 patietns had poorly localized rupture. The median size of VSR was 12 mm (10–17). Median left ventricular (LV) ejection fraction was 40% (IQR: 35–40%).Right ventricular (RV) dysfunction based on TAPSE (tricuspid annular plane systolic excursion) less than 16mm was found in 28 (41.8%) patients. Median Pulmonary artery systolic pressure (PASP) was 50 (40–60) mm Hg. Moderate to severe mitral regurgitation was

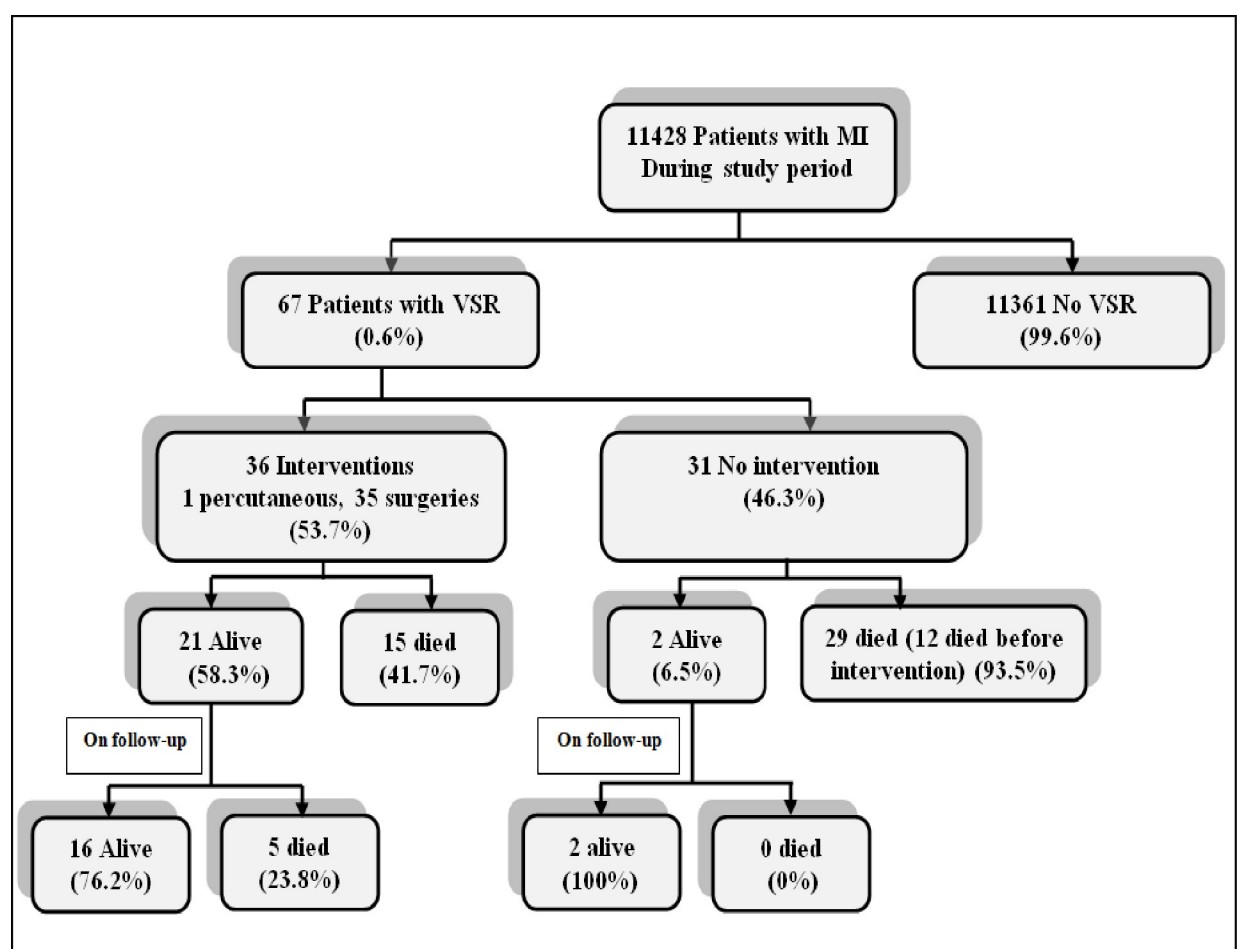

**Fig 1. Flowchart of VSR patients.**

present in 10 (14.9%) patients and 29 (43.2%) had moderate to severe tricuspid regurgitation. Coronary angiography was performed in 51 (76.1%) out of whichsingle vessel disease was seen in 17 (33.3%), and two and three-vessel disease was present in 25.5% and 41.2%, respectively (Table 1).

## 3.2. Management

VSR closure was attempted in 36 patients- surgery in 35 (52.5%) and percutaneous device as initial closure option in one patient. Additional 12 patients were accepted for surgery; however, they rapidly deteriorated and died within 12 hours of VSR diagnosis before surgery could be carried out. In the remaining 19 patients, 11 patients were assessed to be prohibitive risk and declined for surgery by cardiac surgeons, five patients were considered poor candidates for surgery by the treating cardiologist, and in the remaining three patients, families declined surgery. The median mortality risk calculated using Euro Score II was 13.1% in the patients undergoing surgery (IQR: 6–25.1). Only 4(11.4%) patients underwent elective surgery (median time from MI to surgery of 43 days (IQR: 35–54)) while rest had urgent, emergent, or salvage procedures (median time from MI to surgery of 5 days, (IQR: 2–19)). In addition,21 (60%) were on inotropes, and 16 (45.7%) had Intra aortic balloon pump (IABP) inserted before surgery.

**Table 1. Characteristics of subjects with ventricular septal rupture as per surgical and non-surgical intervention (N = 67).**

| Variable | n (%) | Surgical treatment 36(53.7) | Non-surgical Treatment 31(46.3) | p-value |
|---|---|---|---|---|
| Age in years | 62.7 ± 11.1 | 58.4±9.6 | 67.7±10.7 | <0.001 |
| Males | 42 (62.7) | 26 (72.2) | 16 (51.6) | 0.08 |
| Hypertension | 38 (56.7) | 21 (58.3) | 17 (54.8) | 0.8 |
| Diabetes mellitus | 28 (41.8) | 14 (38.9) | 14 (45.2) | 0.6 |
| Smokers | 10 (14.9) | 6 (16.7) | 4 (12.9) | 0.8 |
| Dyslipidemia | 10 (14.9) | 9 (25.0) | 1 (3.2) | 0.01 |
| Prior PCI | 2 (3.0) | 1 (2.8) | 1 (3.2) | 0.9 |
| Prior CABG | 2 (3.0) | 1 (2.8) | 1 (3.2) | 0.9 |
| Prior CVA | 4 (6.0) | 0 (0) | 4 (12.9) | 0.02 |
| Prior CKD | 3 (4.5)0.4 | 1 (2.8) | 2 (6.5) | 0.4 |
| Cardiogenic Shock Class (at time of VSR diagnosis)$ | | | | |
| A & B | 15 (22.4) | 11 (30.6) | 4 (12.9) | 0.08 |
| C,D& E | 52 (77.6) | 25 (69.4) | 27 (87.1) | |
| Intra-Aortic Balloon Pump | 31 (46.3) | 22 (61.1) | 9 (29.0) | 0.009 |
| Initial reperfusion therapy | | | | |
| Any reperfusion (thrombolytic/angioplasty) | 23 (34.3) | 15 (41.7) | 8 (25.8) | 0.6 |
| No reperfusion | 44 (65.7) | 21(58.3) | 23 (74.2) | |
| Median time between MI and VSR diagnosis (days) * | 2 (1–5) | 3 (1.5–11) | 1 (1–3) | 0.003 |
| Median VSR size (mm) * | 12 (10–17) | 13.5 (10–22) | 12 (9.5–16) | 0.4 |
| Basal VSR | 19 (29.2) | 11(30.6) | 8 (25.8) | 0.5 |
| Left ventricular ejection Fraction | | | | |
| ≥ 45% | 16 (23.9) | 9 (25.0) | 7 (22.6) | 0.6 |
| <45% | 51 (76.1) | 27 (75.0) | 24 (77.4) | |
| Right ventricular dysfunction (n = 58)# | 28 (48.3) | 15 (46.9) | 13 (50.0) | 0.8 |
| Mitral regurgitation (n = 60) | | | | |
| Mild | 50 (83.3) | 29 (82.8) | 21 (84.0) | 0.6 |
| Moderate to severe | 10 (16.7) | 6 (16.7) | 4 (16.0) | |
| Median pulmonary artery systolic pressure (mmHg)* | 50 (40–60) | 50 (35–55) | 50 (35–55) | 0.9 |

Values are means ± SD for continuous variables

* median with interquartile range wherever stated, and n (%) for categorical variables, MI: Myocardial Infarction, VSR: Ventricular septal rupture, $ SCAI classification of cardiogenic shock.

# TAPSE (Tricuspid annual place systolic excursion) was used to assess right ventricular fuction.

### 3.3. Surgical technique

The surgical technique used on patients in our settings was similar to one popularized by Komeda and David et al. in 1990 [16]. This was via median sternotomy, aortic and bicaval cannulation, aortic cross-clamping, antegrade blood cardioplegia, and hypothermic cardiopulmonary bypass in all patients. VSR was approached through a left ventriculotomy in most patients, and the infarct exclusion technique was utilized using a single bovine pericardial patch with a running suture to healthy myocardium. Some of the basal and posterior VSRs were approached via the right ventricle. No surgical adhesives were used. Ventriculotomy was closed using interrupted sutures secured by Teflon felt pledgets.

Concomitant CABG was performed in 21 (60%). The median cardiopulmonary bypass time was 85 minutes (IQR: 57–109). Post-operative acute kidney injury occurred in 16 (45.7%) patients, and one patient required hemodialysis. 28 patients (80.0%) required transfusion of more than 2 packs of red blood cells. Residual VSR likely due to patch dehiscence was present

on transthoracic echo in 9 of the 35 (25%) of patients who underwent surgery. One additional patient underwent percutaneous device closure three weeks after failed surgical VSR closure due to New York Heart Association (NYHA) IV heart failure symptoms. Both patients with percutaneous device closure died during the hospital stay.

### 3.4. Mortality

During the hospital stay, 44 (65.7%) patients died, 29 of 31 (93.5%) patients who did not undergo surgery, and 15 of 36 patients who underwent intervention. The median follow-up was 44.1 months (IQR-11.6–73.4) in the 23 patients discharged alive. There were 5 deaths during the follow-up. One death was within 28 days of discharge due to patch dehiscence and failed rescue percutaneous device closure; another patient died five years post-discharge due to right heart failure. Rest of three deaths were due to COVID, renal failure, and stroke.

The incidence rate (IR) of mortality in overall VSR patients was 6/100 person-months (95% CI: 4-8/100 person-months) (Fig 2), 89/100 person-months (95% CI:62-129/100 person-months) in patients who didn't undergo surgery, and 2/100 person-months (95% CI:2-4/100 person-months)in patients who underwent surgical treatment. The log-rank test indicated a significant difference in survival of patients with and without surgical intervention for VSR (p-value<0.001).

### 3.5. Health-related quality of life (HRQOL)

HRQOL was evaluated in 11 patients. The mean overall HRQOL score on KCCQ-12 was 73.9 ±9.3. The median scores of physical limitations, symptoms, QOL, and social limitations

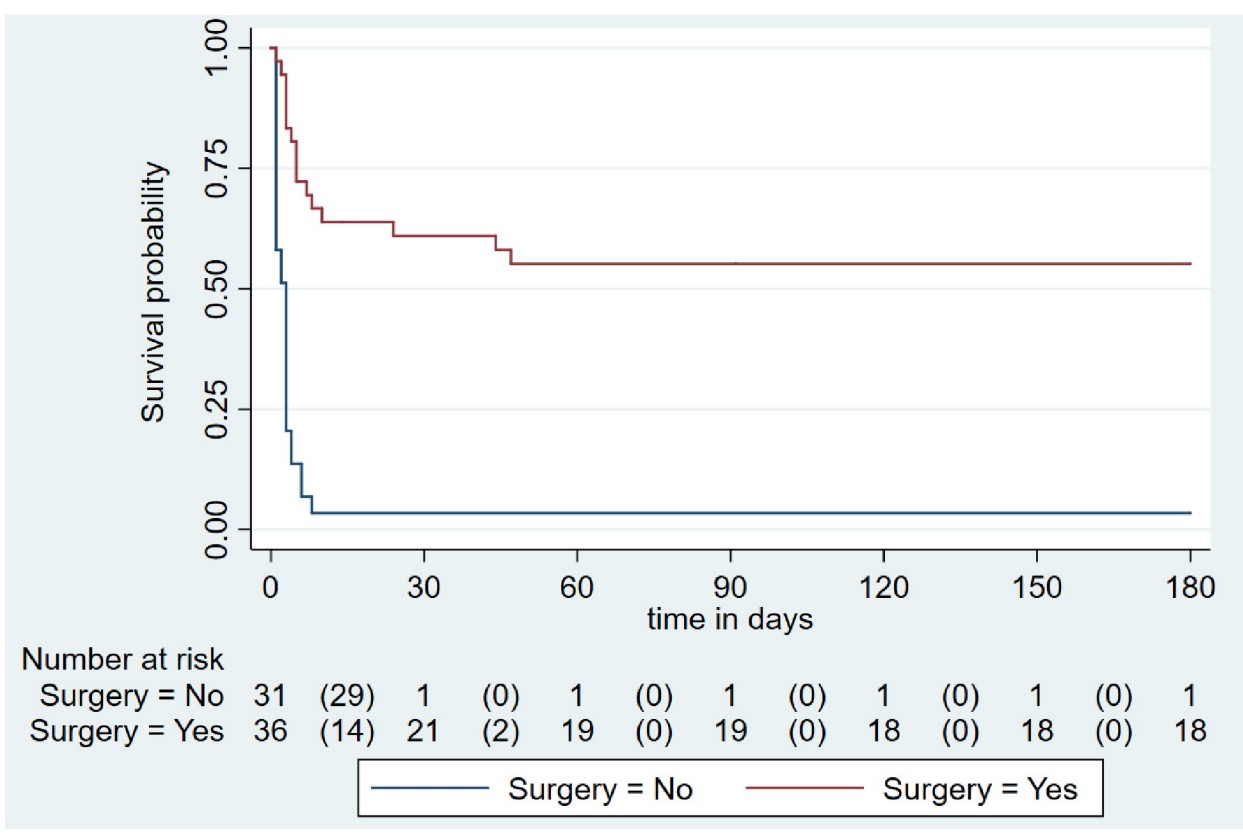

**Fig 2. Kaplan-Meier survival curve of operated and not operated VSR patients (n = 67).**

subscales were 66.6 (IQR 53.3–80), 85 (IQR 80–95), 75 (IQR 62.5–75), and 80 (IQR 60–80). As per the cutoffs, 54.5% (n = 6) had good overall HRQOL, and 45.5% (n = 5) had excellent overall HRQOL.

## 3.6. Predictors of in-hospital mortality

In univariate analysis, hospital mortality was significantly associated with non-surgical management, basal VSR, RV dysfunction, early VSR post-MI, and severe cardiogenic shock at admission (class C, D, or E). There was no significant association of cardiovascular risk factors, Killip class, SBP, or heart rate to mortality. In addition, no significant statistical association was found between in-hospital mortality and absence of reperfusion, multi-vessel vs. single-vessel CAD, defect size, PASP, or LV function. In multivariable analysis, basal VSR location, advanced stages of cardiogenic shock (class C, D, or E), and non-surgical management were significantly associated with hospital mortality (Table 2).

**Table 2. Univariate and multivariable analysis of covariates for hospital mortality$^\$$.**

| Parameters | Total n(%) | Survivors 23(34.3) n (%) | Non-survivors 44(65.7) n(%) | Crude Odds ratio (95% CI) | p-value | Multivariable analysis p-value |
|---|---|---|---|---|---|---|
| OVERALL | | | | | | |
| Age | 62.7 ± 11.1 | 59.1±11.0 | 64.6±10.8 | 1.05 (0.9–1.1) | 0.06 | 0.6 |
| Female gender | 25(37.3) | 6(26.1) | 19(43.2) | 2.1(0.7–6.5) | 0.2 | 0.4 |
| Diabetes | 28 (41.8) | 6(26.1)) | 22(50.0) | 2.8(0.9–8.5) | 0.06 | 0.3 |
| VSR within 1 week of MI | 54(80.6) | 15(65.2) | 39(88.6) | 4.2(1.2–14.7) | 0.02 | 0.4 |
| Basal VSR location | 19 (29.2) | 3(13.0) | 16(36.4) | 3.6(1.1–11.4) | 0.03 | 0.02 |
| Cardiogenic shock (at time of VSR diagnosis)* | | | | | | |
| Class A & B | 15(22.4) | 12(52.2) | 3(6.8) | 1 | 0.002 | 0.006 |
| Class C, D & E | 52(77.6) | 11(47.8) | 4193.2) | 15.4(1.9–139.3) | | |
| Non-surgical management for VSR | 31(46.3) | 2(8.7) | 29(65.9) | 20.4(4.2–100) | <0.001 | <0.001 |
| Right ventricular dysfunction | 28 (48.3) | 4(17.4) | 24(54.5) | 7.8(2.2–28.2) | 0.001 | 0.09 |
| Moderate/severe mitral regurgitation | 10 (14.9) | 3(13.0) | 7(15.9) | 1.3(0.3–5.4) | 0.7 | 0.8 |
| IABP | 31 (46.3) | 10(43.5) | 21(47.7) | 1.2(0.4–3.2) | 0.8 | 0.5 |
| INTERVENTION SUBGROUP | | | | | | |
| Parameters | Total 36 (53.7) | Survivors 21(58.3) n(%) | Non-survivors 15(41.7) n(%) | Crude Odds ratio (95% CI) | p-value | Multivariable analysis p-value |
| Age | 58.4±9.6 | 58.8±10.8 | 57.9±10.8 | 1.0(0.9–1.06) | 0.7 | 0.5 |
| Female gender | 10(27.8) | 5(23.8) | 5(33.3) | 1.6(0.4–6.9) | 0.5 | 0.3 |
| Basal VSR location | 11(57.9) | 3(14.3) | 8(53.3) | 6.4(1.4–28.6) | 0.01 | 0.1 |
| Cardiogenic shock (at time of surgery)* | | | | | | |
| Class A & B | 12(33.3) | 11(52.4) | 1(6.7) | 1 | 0.004 | 0.04 |
| Class C, D & E | 24(66.7) | 10(47.6) | 14(93.3) | 15.4(1.7–139.2) | | |
| Surgery within 7 days of MI | 17(47.2) | 7(33.3) | 10(66.7) | 4.0(0.98–16.3) | 0.048 | 0.2 |
| Right ventricular dysfunction | 15(53.6) | 4(19.0) | 11(73.3) | 20.6(3.2–133.4) | 0.001 | 0.004 |
| Emergent or salvage procedure | 18(50) | 8(38.1) | 10(66.7) | 3.3(0.8–13.0) | 0.09 | 0.4 |
| Pre-operative IABP insertion | 16(44.4) | 6(28.6) | 10(66.7) | 5.0(1.2–3.2) | 0.04 | 0.6 |
| Concomitant CABG | 21(60) | 13(61.9) | 8(53.3) | 0.8(0.2–3.2) | 0.7 | 0.4 |

$^\$$ Variables were added into models based on clinical significance and/or univariable p-values.

MI: Myocardial Infarction, VSR: Ventricular septal rupture, CABG: Coronary arteries bypass grafting IABP: Intra aortic balloon pump.

* SCAI classification of cardiogenic shock.

In patients who underwent intervention, basal VSR, RV dysfunction, surgery within 7 days post- MI, preoperative IABP, and severe shock (class C, D, or E) at the time of surgery correlated with increased mortality. There was no association between concomitant CABG surgery and emergent or urgent procedure indication and hospital outcomes. In multivariable analysis, advanced stages of cardiogenic shock (class C, D, or E) and RV dysfunction were significantly associated with hospital mortality (Table 2).

## 4. Discussion

In this study, we looked at the clinical characteristics, management patterns, in-hospital and intermediate-term outcomes of an all-comers population of patients presenting with VSR after myocardial infarction to a single specialty facility in a Low-Middle-Income Country (LMIC). These patients have a very high early mortality. Infarct location, early VSR post MI, worse severity of the cardiogenic shock, right ventricular dysfunction, and non-selection for surgery were predictors of early mortality. The small number of survivors available for follow-up had a good intermediate-term survival with an acceptable HRQOL.

We found a rate of VSR over 10 years of 0.6%. This is comparatively lower than the 1–3% VSR rate reported in studies conducted in the pre-reperfusion era [17]. In a study conducted in India, the rate of VSR was comparable to our findings (0.7%) [18]. However, the incidence of VSR in the GUSTO-1(Global Utilization of Streptokinase and TPA for Occluded Coronary Arteries) trial was slightly lower (0.2–0.4%) as compared to our findings [19]. Timely reperfusion therapy reduces infarction size and subsequent rupture complications [20, 21]. The higher VSR rate in our study is likely due to the inclusion of all comers' myocardial infarction population compared to many reports that only studied patients receiving early reperfusion therapy. Two-thirds of patients in our study received no reperfusion therapy due to late presentation to the health care facility. VSR patients were also referred from other hospitals for further management, increasing the frequency. At the same time, underestimation due to rapid deterioration and death before the diagnosis of VSR is established, cannot be excluded.

Our in-hospital VSR related mortality rate was 65.7%. However, a study conducted in similar resource-limited settings on post-MI VSR patients suggested a higher rate (80.4%) of in-hospital mortality [18]. In contrast, findings from developed countries exhibited comparatively lower rates of early VSR related mortality ranging from 35%-54% [17, 22, 23]. The difference in the rate of in-hospital mortality between our and other reports is likely due to the difference in high-risk features in the included patients as well as the proportion who underwent surgery. Our study also showed a very dismal prognosis of patients who did not undergo surgery which is consistent with prior reports. In the SHOCK trial registry and GUSTO-I trial, patients with VSR who did not undergo surgery, the mortality was 94–95%. Mortality after surgery in the SHOCK registry was 87% and 47% in the GUSTO I trial [19]. In our study, the patients who underwent surgical repair, the mortality rate is 41.6% and comparable to the mortality rates demonstrated in other studies from developed countries[17, 23]. The largest report of 2876 patients with VSR undergoing surgery in 2012 from the database of The Society of Thoracic Surgeons showed overall operative mortality of 42.9% and a 54.1% mortality in patients undergoing surgery within 7 days from MI [24], nearly identical to the surgical outcomes in our report.

The presence of cardiogenic shock has been a consistent predictor of mortality in all published reports among patients with VSR treated with or without surgical or percutaneous closure [5, 19]. We have shown the application of the recently published SCAI shock classification and found advanced cardiogenic shock to correlate with mortality. Overall, 77% of the patients in our study had advanced stages of cardiogenic shock, class C, D, or E as per SCAI shock classification with a 78.9% mortality vs. 20% in patients with class A or B shock.

Thiele et al., GUSTO 1 trial, SHOCK registry, and outcomes from the Society of Thoracic Surgeons national database also show very high mortality in VSR patients presenting with cardiogenic shock [5, 19, 24, 25]. We also found inferior infarction that is usually associated with basal VSR location and right ventricular dysfunction, early VSR post MI, and early surgery as predictors of mortality [26, 27]. All these are consistent with published literature. In autopsy studies [28], inferior infarcts and basal VSR location are anatomically more complex and surgically more challenging as these are usually associated with intramyocardial dissection and lead to multiple hemorrhagic tracts with myocardial disruption and necrosis extending beyond primary infarct location. There is more extensive interventricular septal and right ventricular dysfunction, which might also explain the higher rates of in-hospital mortality in such patients [17]. Anterior infarcts are usually apically located and associated with localized through and through defects that are easier to surgically approach and repair. Though statistically not significant, likely due to small sample size, gender may play a role in survival since women are less likely to be sent for surgical intervention (40 vs. 61.9%) and have poor survival (24% vs 40.4%) when compared to men. Unlike in previous studies, advanced age was not found to be significantly associated with in-hospital mortality [18].

In the present study, the timing from MI onset to VSR diagnosis and MI to surgery appears longer for patients who survived after surgery. 77.7% of patients who underwent surgery after 7 days survived compared to 41.2% in patients whose surgery was performed within 7 days of myocardial infarction. This has also been reported in most series published so far [29, 30]. However, this is likely explained by survivorship bias. Of our patient group, a large (80.6%) majority developed VSR within 1 week of myocardial infarction diagnosis, most (77.6%) presented with advanced shock, with overall very poor outcome including the 17.9% who died while waiting for surgery. In view of such high early mortality, delaying surgery may not improve the outcome of this very high-risk patient group. More aggressive clinical and echocardiographic evaluation of at-risk patients with a view towards earlier diagnosis and surgery before shock state gets established may lead to improved outcomes. Once the shock is established, use of advanced mechanical circulatory support (MCS) devices, used alone or in combination, such as IABP, Impella, veno-arterial extracorporeal membrane oxygenation(V-A ECMO), and left ventricular assist device (LVAD) to stabilize these patients and proceeding with surgery at a later time when more myocardial healing has occurred, may be considered [31, 32]. The surgical challenges in these patients of bleeding at the ventriculotomy site, difficulty in secure fixation of the patch with subsequent increased dehiscence and recurrence of the shunt, and low output syndrome may potentially decrease if surgery is performed later. There have been some recent single-center case series using MCS strategies with lower early mortality [33]. It is difficult to make any conclusions from these limited series, although there will likely never be any adequately powered randomized trials or even large case series of such patient's and in view of dismal early outcomes with medical stabilization attempts, use of MCS to stabilize the patients before surgery would seem to be the most clinically appropriate approach. However, most of the MCS devices have very limited availability even in the advanced health care system, are generally cost-prohibitive and the availability of emergency VA-ECMO teams is restricted to select centers. At our institution only IABP is available. These patients should ideally be treated at a center with available resources and a heart team comprising of the cardiac intensivist, interventional cardiologist, cardiac surgeons, and cardiac anesthesia to achieve the best outcomes for these patients. Some patients did not undergo surgery due to surgeons or physicians' subjective assessment of extreme age or limited functional capacity. Frailty assessment at times can be a very difficult task; however, the heart team can use frailty index systems such as Fried Frailty Index (FFI) or other similar indices to do this assessment more objectively [34].

The most concerning finding of our study is that majority (65.7%) of the patients in our study did not receive reperfusion therapy due to late presentation to the hospital. Even the patients who received reperfusion with thrombolysis or PCI, likely did not receive timely treatment or achieve tissue level perfusion resulting in large areas of myocardial necrosis [35]. There is a need to develop community-level awareness of myocardial infarction, ECG availability, ambulance systems with ECG acquisition, transmission and thrombolytic administration capability and establishing hub and spoke model STEMI networks with thrombolytic administration and PCI capability [36]. In resource constraint economies, a balance needs to be maintained between thrombolytic therapy with early cardiac catheterization and PCI when required and acute STEMI PCI with focus on very early recognition and reperfusion treatment for all eligible STEMI patients. In the majority of patients presenting beyond 12 hours, PCI still leads to significant myocardial salvage and reduction in infarct size [37, 38]. Immediate PCI should be considered in all patients presenting up to 24 hours after symptom onset and in patients with ongoing chest and ST elevation on ECG, even up to 48 hours after symptoms onset [39, 40].

So far, very limited data is available on VSR associated mortality and its influencing factors in LMICs. Therefore, our study supplements the findings from resource-limited countries. Moreover, this study also determined survival outcomes and characteristics of patients who didn't undergo surgery, which is rarely explored in literature. Classification of stages of cardiogenic shock as per SCAI guidelines to rationalize the mortality rates in VSR was another strength of this study as limited literature is available using stages of cardiogenic shock as a predictor of Post-MI-VSR mortality. We also evaluated HRQOL in the surviving patients which have not been reported so far in this patient group.

This study presents certain limitations. Limited sample size and inadequate statistical power were some of the major limitations of this study. Furthermore, in 14 patients, immediate death ensued before any further treatment was done; hence further details couldn't be extracted. Lastly, KCCQ-12 wasn't validated or previously used in the Pakistani population and in VSR patients.

## 5. Conclusion

VSR is a rare post-MI complication with very high mortality rates. Surgical treatment remains one of the foremost options in treating VSR for better survival outcomes. However, the timing of surgery and advanced stages of shock at presentation also play a crucial role in determining the survival of VSR patients. Therefore, we recommend SCAI shock classification for triaging patients for further management in such a high-risk population.

## Acknowledgments

We would like to thank Mr.Imran Ali for helping us with the surgical data extraction. We would also like to extend our gratitude to all the patients and their families for permitting us to use their data and cooperating with us on the telephonic interviews. We would also like to thank Dr. Madiha Ahmed for her assistance in collecting quality of life information from the patients.

## Author Contributions

**Conceptualization:** Asad Pathan.

**Data curation:** Marium Mustaqeem, Sharwan Bhuro Mal.

**Formal analysis:** Saba Aijaz, Ghazal Peerwani, Sana Sheikh.

**Investigation:** Asad Pathan.

**Methodology:** Saba Aijaz, Ahson Memon, Ghufranullah Khan, Asad Pathan.

**Software:** Saba Aijaz, Ghazal Peerwani.

**Supervision:** Ahson Memon, Ghufranullah Khan, Asad Pathan.

**Writing – original draft:** Saba Aijaz, Ghazal Peerwani, Asadullah Bugti, Asad Pathan.

**Writing – review & editing:** Saba Aijaz, Ghazal Peerwani, Sana Sheikh, Asad Pathan.

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
