## [Decision Letter · Decision Letter 0]

31 Jan 2022

PONE-D-21-37769Management and outcome of post-myocardial infarction ventricular septal rupture- a Low-Middle-Income Country ExperiencePLOS ONE

Dear Dr. Aijaz,

Thank you for submitting your manuscript to PLOS ONE. After careful consideration, we feel that it has merit but does not fully meet PLOS ONE’s publication criteria as it currently stands. Therefore, we invite you to submit a revised version of the manuscript that addresses the points raised during the review process.

We look forward to receiving your revised manuscript.

Kind regards,

Jeffrey J. Rade, MD

Academic Editor

PLOS ONE

Journal Requirements:

Reviewers' comments:

Reviewer's Responses to Questions

**Comments to the Author**

1. Is the manuscript technically sound, and do the data support the conclusions?

Reviewer #1: Partly

Reviewer #2: Partly

2. Has the statistical analysis been performed appropriately and rigorously? 

Reviewer #1: Yes

Reviewer #2: I Don't Know

3. Have the authors made all data underlying the findings in their manuscript fully available?

Reviewer #1: No

Reviewer #2: Yes

4. Is the manuscript presented in an intelligible fashion and written in standard English?

Reviewer #1: No

Reviewer #2: Yes

5. Review Comments to the Author

Reviewer #1: In this manuscript titled “Management and outcome of post- myocardial infarction ventricular septal rupture: a Low- Middle- Income Country Experience”, the authors reviewed patients presenting with delayed acute myocardial infarction (AMI) and complicated by ventricular septal rupture (VSR) in a single center in a low- middle income country. Authors examined patients’ clinical and demographic characteristics, presentation, management, in-hospital and 1-year mortality in this patients’ population.

- Major Issues:

o Patients’ characteristics, Presentation, Risk factors: Paper mentions multiple significant risk factors for VSR in ACS patients that have been established in the literature such as: history of CVA/ prior strokes, cardiac arrest, ST elevation on initial ECG, chronic kidney dysfunction (CKD), positive cardiac biomarkers; however, this paper reports only hx of HTN and DM, and neglect to mentions other risk factors/ patients’ feature that could be pertinent such as the following:

Whether patients had chest pain on presentation?

Initial EKG showing ST elevation? Q waves?

Baseline kidney function or creatinine on presentation?

Troponin or CK-MB level on presentation.

Prior history of strokes/ CVA?

Prior history of MI/ or prior PCI/CABG?

Did the patient receive heparin or P2Y12 on presentation or during hospitalization?

Timing of tPA or PCI if given?

o RV dysfunction was stated in the paper as a predictor of in-hospital mortality; however, the paper does not mention the parameters used to define RV dysfunction such as: RV size, fractional area change, TAPSE on transthoracic echocardiogram or other hemodynamic data from right heart catheterization.

o The paper claims that the “study determines survival outcomes and characteristics of patients who survived ACS complicated by VSR without surgical intervention” (Line 351-353).

Flow chart/ results section from the paper showed 2 patients out of 31 who survived ACS complicated by VSR without intervention. This is such a small number to draw any conclusion regarding characteristics and risk factors.

o 14/67 patients expired on arrival to the hospital thus cannot use this population to draw concrete conclusions.

o Euroscore II was reported on patients accepted for surgical intervention. It would be helpful to report the STS score or the Euroscore II on patients declined for surgery to be able to objectively compare the surgical risk of these two groups.

- Minor issues:

o Missing reference (lines 76-78).

o There is a lack of consistency with abbreviation use. Authors already introduced MI as the abbreviation for myocardial infarction earlier in the manuscript but used the abbreviation sparingly throughout the manuscript.

o Non- significant p-values are reported as “NS” instead of the true value, I would recommend stating the actual number as it can provide additional information to the reader.

o There are numerous grammatical issues in the paper that require thorough and critical examination.

o Results / conclusion sections are difficult to follow. It may be helpful to present data in a more organized/ group specific fashion to make it easier to follow.

Reviewer #2: Please see attachment for full review.

The data largely supports the conclusions aside from recommended revisions (elaborated in attached text).

I do not know if the statistical methods were sound due to the fact that the data reported in the tables are presented in an unconventional way (see full comments for further elaboration). I do not know if the tests to assess for statistical significance were conducted appropriately and just the data was presented unconventionally or if the tests to assess for statistical significance were performed incorrectly as well.

6. PLOS authors have the option to publish the peer review history of their article (what does this mean?). If published, this will include your full peer review and any attached files.

Reviewer #1: **Yes: **Summer Aldrugh

Reviewer #2: No

---

## [Author Response · Author response to Decision Letter 0]

1 Aug 2022

Thank you for the valuable feedback and suggestions from the reviewers' feedback. Detailed answers to the reviewers' comments are attached. there was a figure suggested by one of the reviewers. We have attached the figure in the response to the reviewer document. however if needs be to make it part of the original article, we can upload it separately and refer to it in the manuscript.

RESPONSE TO REVIEWERS' COMMENTS

Reviewer #1: 

In this manuscript titled “Management and outcome of post- myocardial infarction ventricular septal rupture: a Low- Middle- Income Country Experience”, the authors reviewed patients presenting with delayed acute myocardial infarction (AMI) and complicated by ventricular septal rupture (VSR) in a single center in a low- middle income country. Authors examined patients’ clinical and demographic characteristics, presentation, management, in-hospital and 1-year mortality in this patients’ population.

-Major Issues:

1. Patients’ characteristics, Presentation, Risk factors: Paper mentions multiple significant risk factors for VSR in ACS patients that have been established in the literature such as: history of CVA/ prior strokes, cardiac arrest, ST elevation on initial ECG, chronic kidney dysfunction (CKD), positive cardiac biomarkers; however, this paper reports only hx of HTN and DM, and neglect to mentions other risk factors/ patients’ feature that could be pertinent such as the following:

a. Whether patients had chest pain on presentation?

We did not give chest pain data since our focus was on VSR where most patients presented with either shock or dyspnea

b. Initial EKG showing ST elevation? Q waves.

We do not have data on ECG details at presentation

c. Baseline kidney function or creatinine on presentation?

We had data on pre-operative acute renal failure which was present in 16/36 patients

d. Troponin or CK-MB level on presentation.

Not available

e. Prior history of strokes/ CVA? 

Data added in table 1, page 7

f. Prior history of MI/ or prior PCI/CABG?

Data added in table 1, page 7

g. Did the patient receive heparin or P2Y12 on presentation or during hospitalization?

All patients received Heparin and P2Y12 inhibitors

h. Timing of tPA or PCI if given? 

Within 24 hours of symptoms

2. RV dysfunction was stated in the paper as a predictor of in-hospital mortality; however, the paper does not mention the parameters used to define RV dysfunction such as: RV size, fractional area change, TAPSE on transthoracic echocardiogram or other hemodynamic data from right heart catheterization.

TAPSE was utilized, page 8, and line 147

3. The paper claims that the “study determines survival outcomes and characteristics of patients who survived ACS complicated by VSR without surgical intervention” (Line 351-353). Flow chart/ results section from the paper showed 2 patients out of 31 who survived ACS complicated by VSR without intervention. This is such a small number to draw any conclusion regarding characteristics and risk factors

We gave details on outcomes of patients who did not undergo surgery. Most of the previous literature only discussed patients who underwent surgery. We did not draw any conclusions on those patients who survived without surgery. Possibly those were natural survivors.

4. 14/67 patients expired on arrival to the hospital thus cannot use this population to draw concrete conclusions. 

Thank you, the point is taken.

5. Euroscore II was reported on patients accepted for surgical intervention. It would be helpful to report the STS score or the Euroscore II on patients declined for surgery to be able to objectively compare the surgical risk of these two groups. 

Euroscore was available for only those patients who underwent surgery so comparison could not be done.

- Minor issues:

1. Missing reference (lines 76-78). 

Line removed

2. There is a lack of consistency with abbreviation use. Authors already introduced MI as the abbreviation for myocardial infarction earlier in the manuscript but used the abbreviation sparingly throughout the manuscript. 

Abbreviations corrected.

3. Non- significant p-values are reported as “NS” instead of the true value, I would recommend stating the actual number as it can provide additional information to the reader. 

P-values added in the table 1and table 2

4. There are numerous grammatical issues in the paper that require thorough and critical examination. 

Language updated

5. Results / conclusion sections are difficult to follow. It may be helpful to present data in a more organized/ group specific fashion to make it easier to follow.

Conclusion wordings rephrased

 

Reviewer #2: 

Please see attachment for full review.

The data largely supports the conclusions aside from recommended revisions (elaborated in attached text).

I do not know if the statistical methods were sound due to the fact that the data reported in the tables are presented in an unconventional way (see full comments for further elaboration). I do not know if the tests to assess for statistical significance were conducted appropriately and just the data was presented unconventionally or if the tests to assess for statistical significance were performed incorrectly as well.

Thank you for the detailed feedback, we have tried our best to improve the tables and data presentation as per the guidance from reviewers and their feedback.

Key Message:

The authors provide an extensive and thorough retrospective case series of a single specialty facility experience of ventricular septal rupture post-MI in a low-middle income country. They provide a much-needed perspective on the management of a rare but often catastrophic post-MI complication which carries an exceptionally high mortality rate, particularly in those who are unable to undergo surgery. They concluded that a SCAI shock classification of C, D, or E at presentation was a strong predictor of mortality and that determining the optimal time for surgical intervention remains a complex problem to solve. Noting that while patients who have delayed surgery have better survival rates, a strong survivorship bias exists. They further included long-term health-related quality-of-life data, attained from survivors. They should be applauded for their hard work. 

-Major Points:

1. Results section 2.9 Predictors of Mortality (page 10, line 200): non-surgical management is referred to as a predictor of hospital mortality. In the review of the clinical characteristics of these 31 patients reviewed elsewhere in the paper – 12 were initially intended to go to surgery but rapidly deteriorated within 12 hours of this decision and died prior to surgery. In another 16, patients were considered prohibitive or poor candidates for surgery by the medical team. In the final 3, the family declined invasive measures. With the above in mind, it was not necessarily the decision to pursue non-surgical management that led to their demise, but rather clinical and demographic characteristics that made them poor or prohibitive candidates for surgery that led to poor clinical outcomes. This should be clarified and elaborated on in the text. 

Clarified in the result section (page 10, line 205-206)

2. From Table 1, 87% of patients in the non-surgically managed patients had class C, D, or E shock and only 29% of all patients in the non-surgically managed group received mechanical support in the form of an IABP. Comparatively, in the surgically managed group 69% had stage C, D, or E shock and 61% of all patients in the surgically managed group received an IABP. Similarly, 42% of patients in the surgically managed group received reperfusion compared to only 26% in the non-surgically managed group. Are you able to tease out exactly what led to decreased use of MCS/ IABP and reperfusion in the non-surgically managed group? Was there some common clinical feature that made patients in this group to be less favorable candidates for MCS, reperfusion, and surgery? Co-morbid conditions? Were there gender differences? Would suspect time-to-presentation to be a driver of this, however per Table 1, the median time between MI and VSR was greater in the surgically managed group compared to the non-surgically managed group. 

Regarding 19/27 patients who were in SCAI shock C, D or E class and were not selected for surgery as well as MCS/IABP due to physician/cardiothoracic surgeon’s decision owing to prohibitive surgery risk. These patients had other comorbids such as acute kidney failure, acidosis, multi-organ dysfunction, cardiac arrest however no frailty criteria was utilized to quantify surgical risk. Mean age was higher in patients who were not intervened surgically while gender had no significant association. 

3. Table 1, page 7: I would report the percentage of patients in the surgical treatment arm that were SCAI shock class A&B and C,D&E rather than the % of SCAI shock A&B patients in surgical treatment arm and so on (the first column of “all patients” is formatted this way while the subgroups are formatted differently). This is the same for rows: “initial reperfusion therapy”, “LV EF”, and “MR”

Percentages have been updated in Table 1, page 7

4. Table 1, page 7: For the rows “IABP” , “Initial reperfusion therapy” , “basal VSR” , and “RV dysfunction” – this table is meant to compare the baseline characteristics of the surgically treated and non-surgically treated groups (the test for statistical significance is to determine if the baseline characteristics of the two groups are different), with this in mind I would report the % of patients within each group that, for example, received a IABP or were revascularized rather than the distribution of patients with an IABP between the two groups. (The cumulative % of each of these rows = 100%, this should not be the case if attempting to compare baseline characteristics between the two groups). 

Percentages have been updated in Table 1, page 7

5. Table 2, page 12: Again, this table is meant to compare the baseline characteristics of the survivors and non-survivors (the test for statistical significance is to determine if the baseline characteristics of the two groups are different), with this in mind I would report the % of patients within each group that, for example, had a VSR within 1 week of MI rather than the distribution of patients who had a VSR within 1 week of an MI between the two groups. The % of patients with a VSR within 1 week is the 2 groups is 65% in the survivors and 88% in the non-survivors, the %s displayed are 28 and 72%, respectively – this is misleading. For another example, when comparing the two groups it is more helpful to know that only 26% of the survivors were female and that 43% of the non-survivors were female. While it is helpful to know that only 24% of females survived compared to 40% of males, it is not the aim of this table. 

Percentages have been updated in Table 2, page 12

6. Discussion (page 15, line 303): While not statistically significant in these 67 patients, only 40% of women went on to surgery compared to 62% of men and only 24% of females survived compared to 40% of males. This suggests that gender may play a role in survival however the patient population was not large enough to see a statistically significant difference.

Comments added; Page 16, Line 308-311

7. Are there data regarding the coronary artery involved or distribution of ST changes on EKG? Did this correlate with the distribution of the VSR?

40 patients (59.7%) had anterior MI and rest had non anterior MI location mentioned in page 6 line 129, since coronary angiography was performed in 51 (76.1%) we did not report coronary distribution. Also we did not gather ECG data.

8. The utility of the KC Questionnaire in determining the value of surgery is a little unclear. It does suggest that appropriate surgical candidates, who undergo surgery and survive have a good subjective quality of life, but it does not necessarily suggest that poor candidates for surgery who have similar QOL. 

Agree to the reviewer on this point, the KC questionnaire was only utilized to see that once patient has had a successful surgical outcome, they tend to do well subjectively in long term as well. 

-Minor Points 

1. Recommend language editing.

Thank you, done

2. Recommend statistical review.

We did statistical methodology review and updated the tables too.

3. Introduction (page 4, line 72): Please clarify whether this is acute heart failure or a history of chronic heart failure. Later refer to high Killip class to be a risk factor for VSR.

Clarification is added in the test

4. Introduction (page 4, line 73): “elevated cardiac biomarkers at the time of presentation” may be clearer that “initial positive cardiac biomarkers.” Done

Clarification dome as “acute heart failure”. Page 4, line 72

5. Introduction (page 4, line 77): is there a reference (e.g. autopsy study) for pre-existing coronary disease leading to increased incidence of fatal rupture?

The line has been removed since it created confusion. The sentence hypothesized preexisting coronary disease may cause fatal rupture resulting in less patients reaching alive to get diagnosed with VSR, at the other end preexisting CAD it may lead to formation of collateral circulation that may reduce VSR incidence.

6. Results (page 6, line 129): please clarify what is meant by the onset of the MI (e.g. the onset of symptoms? First contact with medical personnel?). This is relevant to both timing to presentation and timing to VSR. Further, in the instances in which revascularization was deferred – was this only because of timing? 

It is “time from MI diagnosis in a hospital setting to VSR diagnosis”, term updated on page 7, line 132. Revascularization was deferred at the place of primary presentation due to absence of facility for thrombolysis/PPCI. Two thirds of all patients presented elsewhere first and then were referred to our setup once they deteriorated clinically. 

7. Results, Echo and Angiographic Findings (page 8, line 142): how else was the location of the VSR characterized – there are seemingly 8 additional patients that did not have an anterior or basal VSR. Serpiginous or poorly localized?

Eight patients had poorly localized VSR, edited in page 8, line 145

8. Results, Echo and Angiographic Findings (page 8, line 142, 145): 19 patients corresponds to 28% not 29.2 and 28 patients corresponds to 42% not 48.3% of patients. Would recommend reviewing percentages to ensure consistency. 

Thank you for pointing out, corrections done.

9. Results, Echo and Angiographic Findings (page 8, line 149): These data are not shown in Table 1. 

Skipped from table due to redundancy since it is present in the text of the result.

10. Results, Management (page 8, line 160): It is stated that 16 or 46% of patients in the surgically managed group received an IABP, Table 1 reports 22 or 61% of patients received an IABP. Were some placed after or at the time of surgery? Please clarify.

16% received IABP before surgery as mentioned on page 8, line 163, while all IABP including pre and post op were 22.

11. Results, Surgical Technique (page 9, 179): Earlier in the text you refer to 1 patient who underwent percutaneous closure, in this you refer to ‘both’ patients – was there a patient who underwent surgery who later underwent percutaneous closure as well? Please clarify.

Clarification on page 8 line 155-156, page 9 line 181-182

12. Results, Mortality (page 10, 187): while Figure 2 demonstrates the exceptionally high rate of mortality in the immediate days after diagnosis of VSR, this paragraph is convoluted. Namely, the incident rate of mortality in the surgical intervention group is low because some patients survived for a long time, rather than there being a low mortality rate immediately following surgery. May be worth plotting the KM curve of the non-surgical vs surgically managed group in Figure 2 to illustrate the overwhelmingly high mortality in non-surgically managed patients.

KM curve in added below. We did not added it in the manuscript. However if reviewer feels that it may be useful, it can be added.

13. Results, Predictors of Mortality (page 10, 205): was there a difference in outcomes in patients who received MCS specifically in patients with SCAI stage C, D, or E cardiogenic shock? 

IABP was inserted in only SCAI stage C, D or E patients, as per table 2, IABP insertion had no effect on survival in overall group, however more pre-operative IABP were utilized in non survivors (28.6 vs. 66.7) likely due to worse shock state.

14. Results, Predictors of Mortality (page 10, 208): was there a difference in outcomes in patients who received PCI vs thrombolytic? I.e. was the poorer outcome seen with intervention consistent across patients who received PCI as well as thrombolytic? 

There was no statistically significant difference between mortality in PCI vs thrombolytic (4/7 vs 10/16, p-value 0.8)

15. Discussion (page 13, line 255): Would be cautious to use the terminology of “early presentation” was a predictor of early mortality; it seems rather that “early VSR after MI” was a predictor, not early presentation in general. 

Thank you term changes page 10, line 258

16. Discussion (page 14, line 295): Again, would be cautious to use the terminology of “early presentation” but rather “early VSR after MI” – further would be cautious to say early surgery is a predictor of mortality. This is likely due to survivorship bias, as you explained later in the discussion.

Thank you term changes page 10, line 258

17. Discussion (page 15, line 309): It states a large portion of the patient population (81%) presented early after MI – however revascularization was deferred in 66% of patients due to timing. Please clarify this. 

Sentence clarified, this was development of VSR within 1 week of MI and not early MI presentation

18. Discussion (page 16, line 326): This is hard to follow, please clarify what you be the most clinically appropriate approach e.g. early vs delayed strategy.

Clarification added page 16 line 317-318

19. Discussion (page 17, line 349): Was revascularization attempted in the 24-48 hr range in select patients in this study?

Since late patients included in this study were mostly with clinical diagnosis of VSR, surgery was the first line treatment opted in this study and no percutaneous procedures were advised.

Tables and Figures:

1. Table 1, page 7: Recommend keeping decimal points consistent between mean and SD. 

Thank you for pointing out, changes done

2. Table 1, page 7: If reporting significance in Table 1, would recommend reporting p-values rather than “NS” (alternatively, could delete column of p-values altogether and use footnotes to denote statistically significant differences) – being a retrospective case series of a relatively small number of patients (67), it was not necessarily powered to detect differences in these categories. For example, 62% of men went for surgery compared to only 40% of women. While this may have occurred by chance, it is worth further investigation regardless of the p-value. (E.g. it is possible the women were older? Given that in the non-surgical treatment arm, patients were on average older and there was a higher percentage of women in this group.)

P-values added in tables

---

## [Decision Letter · Decision Letter 1]

19 Sep 2022

PONE-D-21-37769R1Management and outcome of post-myocardial infarction ventricular septal rupture- a Low-Middle-Income Country ExperiencePLOS ONE

Dear Dr. Aijaz,

Thank you for submitting your manuscript to PLOS ONE. After careful consideration, we feel that it has merit but does not fully meet PLOS ONE’s publication criteria as it currently stands. Therefore, we invite you to submit a revised version of the manuscript that addresses the points raised during the review process: The reviewers and Editor agree that Figure 2should be replaced with a Kaplan-Meier plat showing survival of the operative and non-operative groups along with an appropriate figure legend. 

We look forward to receiving your revised manuscript.

Kind regards,

Jeffrey J. Rade, MD

Academic Editor

PLOS ONE

Journal Requirements:

Additional Editor Comments (if provided):

The authors should replace Figure 2 with a Kaplan-Meier plot showing the surgical and non-surgical groups along with an appropriate figure legend.

Reviewers' comments:

Reviewer's Responses to Questions

**Comments to the Author**

1. If the authors have adequately addressed your comments raised in a previous round of review and you feel that this manuscript is now acceptable for publication, you may indicate that here to bypass the “Comments to the Author” section, enter your conflict of interest statement in the “Confidential to Editor” section, and submit your "Accept" recommendation.

Reviewer #2: All comments have been addressed

Reviewer #3: All comments have been addressed

2. Is the manuscript technically sound, and do the data support the conclusions?

Reviewer #2: Yes

Reviewer #3: Yes

3. Has the statistical analysis been performed appropriately and rigorously? 

Reviewer #2: Yes

Reviewer #3: Yes

4. Have the authors made all data underlying the findings in their manuscript fully available?

Reviewer #2: Yes

Reviewer #3: Yes

5. Is the manuscript presented in an intelligible fashion and written in standard English?

Reviewer #2: Yes

Reviewer #3: Yes

6. Review Comments to the Author

Reviewer #2: Overall a robust review of this center's experience with ventricular septal rupture.

Key take-aways are that (1) Many patients are not candidates for intervention due to advanced age, co-morbid conditions, and advanced stage of shock. Females were seemingly also disproportionately in the non-intervention group (not statistically significant, however this review was not powered to assess statistical significance for a demographic characteristic). (2) Those who do not go on to have an intervention have an exceptionally high mortality rate. (3) Those who do go on to have surgery also have a high mortality rate with potential risk factors for mortality being: advanced stage of shock, surgery within 7 days of MI, RV dysfunction, and basal VSR.

I do not think the KM curve, as presented, adds much to the paper because it only includes one arm (maybe all comers?) and the time scale is not as help as say a 30-day of 6-month time scale would be.

Reviewer #3: The changes that the authors made have substantially improved the quality of the manuscript. Two point remains outstanding.

1. Figure 2 is a KM plot of the entire cohort and the text indicates that there were differences between patients undergoing intervention versus those that did not. The authors should replace this figure with a KM plot of the two groups. The plot that they included in the response to the reviewers , however, only has 59 subjects, not 67. The authors need to correct this.

2. The authors should include figure legends, especially for figure 2.

7. PLOS authors have the option to publish the peer review history of their article (what does this mean?). If published, this will include your full peer review and any attached files.

Reviewer #2: No

Reviewer #3: No

---

## [Author Response · Author response to Decision Letter 1]

30 Sep 2022

Thank you for a detailed review that help in improving our paper. Here are the responses for minor revisions as asked.

Additional Editor Comments (if provided) 

1. The authors should replace Figure 2 with a Kaplan-Meier plot showing the surgical and non-surgical groups along with an appropriate figure legend.

Updated Figure 2 has been uploaded with appropriate legend. 

Reviewer #2:

1. I do not think the KM curve, as presented, adds much to the paper because it only includes one arm (maybe all comers?) and the time scale is not as help as say a 30-day of 6-month time scale would be.

The KM curve is reconstructed using 2 groups with revised scale, updated figure 2 is uploaded.

Reviewer #3:

1. Figure 2 is a KM plot of the entire cohort and the text indicates that there were differences between patients undergoing intervention versus those that did not. The authors should replace this figure with a KM plot of the two groups. The plot that they included in the response to the reviewers, however, only has 59 subjects, not 67. The authors need to correct this.

The KM curve is reconstructed using 2 groups with revised scale, new figure 2 is uploaded with N=67.

2. The authors should include figure legends, especially for figure 2.

Figure legends are added to the manuscript.

---

## [Editor Report · Decision Letter 2]

11 Oct 2022

Management and outcome of post-myocardial infarction ventricular septal rupture- a Low-Middle-Income Country Experience

PONE-D-21-37769R2

Dear Dr. Aijaz,

We’re pleased to inform you that your manuscript has been judged scientifically suitable for publication and will be formally accepted for publication once it meets all outstanding technical requirements.

Kind regards,

Jeffrey J. Rade, MD

Academic Editor

PLOS ONE
---

## [Editor Report · Acceptance letter]

18 Oct 2022

PONE-D-21-37769R2 

Management and outcome of post-myocardial infarction ventricular septal rupture- a Low-Middle-Income Country Experience 

Dear Dr. Aijaz:

I'm pleased to inform you that your manuscript has been deemed suitable for publication in PLOS ONE. Congratulations! Your manuscript is now with our production department. 

Kind regards, 

on behalf of

Dr. Jeffrey J. Rade 

Academic Editor

PLOS ONE